# Exploring the Functional Relationship between *y*-Type Thioredoxins and 2-Cys Peroxiredoxins in *Arabidopsis* Chloroplasts

**DOI:** 10.3390/antiox9111072

**Published:** 2020-10-31

**Authors:** Ana Jurado-Flores, Víctor Delgado-Requerey, Alicia Gálvez-Ramírez, Leonor Puerto-Galán, Juan Manuel Pérez-Ruiz, Francisco Javier Cejudo

**Affiliations:** Instituto de Bioquímica Vegetal y Fotosíntesis, Universidad de Sevilla and Consejo Superior de Investigaciones Científicas, 41092 Sevilla, Spain; ana.jurado@ibvf.csic.es (A.J.-F.); victor.delgado@ibvf.csic.es (V.D.-R.); alicia.galvez@ibvf.csic.es (A.G.-R.); b82pugal@gmail.com (L.P.-G.)

**Keywords:** 2-Cys peroxiredoxin, thioredoxin *y*, NTRC, thioredoxin *x*, redox regulation, stress response, photosynthesis, chloroplast

## Abstract

Thioredoxins (Trxs) are small, ubiquitous enzymes that catalyze disulphide–dithiol interchange in target enzymes. The large set of chloroplast Trxs, including *f*, *m*, *x* and *y* subtypes, use reducing equivalents fueled by photoreduced ferredoxin (Fdx) for fine-tuning photosynthetic performance and metabolism through the control of the activity of redox-sensitive proteins. Although biochemical analyses suggested functional diversity of chloroplast Trxs, genetic studies have established that deficiency in a particular Trx subtype has subtle phenotypic effects, leading to the proposal that the Trx isoforms are functionally redundant. In addition, chloroplasts contain an NADPH-dependent Trx reductase with a joint Trx domain, termed NTRC. Interestingly, *Arabidopsis* mutants combining the deficiencies of *x-* or *f-*type Trxs and NTRC display very severe growth inhibition phenotypes, which are partially rescued by decreased levels of 2-Cys peroxiredoxins (Prxs). These findings indicate that the reducing capacity of Trxs *f* and *x* is modulated by the redox balance of 2-Cys Prxs, which is controlled by NTRC. In this study, we explored whether NTRC acts as a master regulator of the pool of chloroplast Trxs by analyzing its functional relationship with Trxs *y*. While Trx *y* interacts with 2-Cys Prxs in vitro and in planta, the analysis of *Arabidopsis* mutants devoid of NTRC and Trxs *y* suggests that Trxs *y* have only a minor effect, if any, on the redox state of 2-Cys Prxs.

## 1. Introduction

Plant chloroplasts have the primary function of synthesizing metabolic intermediates that confer photoautotrophy to these organisms, but these organelles have also developed the ability to sense and respond to the environment, which is a crucial function given the sessile lifestyle of plants. Thus, metabolism within chloroplasts is tightly controlled and is coordinated with plant growth and development [1]. The plasticity of photosynthetic metabolism relies on mechanisms that permit the rapid and reversible control of regulated enzymes in metabolic pathways. One of them is based on redox regulation of enzyme activity through thiol–disulfide exchange reactions involving conserved cysteine residues, a universal strategy conserved between kingdoms [2]. However, redox control in heterotrophic and photoautotrophic organisms shows substantially different features. First, in plant chloroplasts, the reducing power required for redox regulation is provided by photo-reduced ferredoxin (Fdx), whereas in heterotrophs it is provided by NADPH. Thus, reducing equivalents are transferred via a Fdx- or a NADPH-dependent thioredoxin (Trx) reductase, namely FTR and NTR, respectively [3]. Secondly, non-photosynthetic organisms possess at most three Trx-encoding genes, whereas chloroplasts are equipped with a large set of Trxs, including the types *x*, *f*, *m*, *y* and *z*, as well as several Trx-like proteins [4,5,6]. The plastidial Trx subtypes are reduced by FTR in vitro [7], in agreement with the notion that the redox regulation within chloroplast is linked to light, an important environmental cue influencing plant development. Accordingly, the function of Trx target enzymes, normally active during the day and inactive in the night, is reversibly controlled in response to light and darkness. Based on their selectivity for reducing target enzymes in vitro, different functions have been assigned to these Trxs. The subtypes *f* and *m* were proposed to regulate metabolic pathways such as carbon assimilation by the Calvin–Benson cycle, whereas Trxs *x* and *y* were proposed to participate in the control of enzymes involved in the antioxidant response [4,5,6]. On the other hand, Trx *z* is a component of the plastid transcriptional machinery and hence is involved in plastid gene expression [8]. Although Trx subtypes are differently abundant within the organelle [9,10] and appear to have functional specificity, genetic studies have shown that plants devoid of individual Trxs, with the exception of Trx *z* [11], display subtle mutant phenotypes [12,13,14,15]. These observations suggest that remaining redox systems may compensate for the function of a specific Trx, highlighting the partial redundancy among chloroplast Trx subtypes. 

The chloroplast Trx redox system is essential for plant growth, as mutants devoid of the catalytic subunit of FTR, the enzyme that fuels reducing equivalents for Trxs, display a lethal phenotype [16]. Nonetheless, the complexity of redox regulation within the organelle is further increased by the presence of a peculiar NTR with a joint Trx domain, termed NTRC, exclusive of organisms that perform oxygenic photosynthesis [17,18,19]. NTRC shows high affinity for NADPH [20] and efficiently reduces thiol peroxidases such as 2-Cys peroxiredoxins (Prxs) [21,22,23]. Additionally, several studies (reviewed in [5,6,24]) link NTRC to chloroplast enzymes involved in a variety of processes regulated by Trxs, thus indicating that NTRC and Trx targets overlap to some extent. In line with this possibility, NTRC interacts with different chloroplast proteins, most of them previously shown to be regulated by Trxs [25,26,27,28], thus suggesting the concerted action of NTRC and Trxs in the regulation of these targets. Moreover, *Arabidopsis* mutants combining the lack of NTRC with that of Trxs *f* or *x* show very severe growth inhibition phenotypes [29,30], implying that the activity of a specific Trx subtype becomes more relevant in the absence of NTRC. In support of this notion, the combination of decreased levels of FTR and depletion of NTRC leads to a lethal phenotype [27], indicating that both redox systems are required for autotrophic growth. Finally, the finding that the overexpression of an NTRC variant with an inactive Trx domain partially complemented the phenotype of the *ntrc* mutant suggest that this enzyme could interact with plastid Trxs through its NTR domain [26]. 

We have recently shown that the function of the chloroplast redox systems, Trxs and NTRC, is integrated by the redox balance of 2-Cys Prxs [31]. While most Trx subtypes and Trx-like proteins are capable of reducing 2-Cys Prxs in vitro [22,27,32,33,34,35,36], NTRC is the primary reductant of these enzymes in vivo [12], thereby minimizing the draining of electrons from Trxs to 2-Cys Prxs. The absence of NTRC would increase drainage of reducing equivalents from Trxs to 2-Cys Prxs, impairing the redox regulation of their targets. In support of this notion, the severe growth inhibition phenotypes of NTRC-deficient plants are partially rescued by decreased levels of 2-Cys Prxs [31,37]. Therefore, the activity of NTRC maintains the redox state of 2-Cys Prxs, hence allowing light-dependent redox regulation of Trx targets. During the night, 2-Cys Prxs participate in the oxidative inactivation of Trx-regulated enzymes by transferring reducing equivalents from reduced proteins to hydrogen peroxide, Trxs acting as intermediates between the reduced targets and 2-Cys Prxs [36,38,39]. 

Current knowledge suggests that the NTRC-2-Cys Prx system controls the activity of Trxs *f* [31] and *x* [37]. In this regard, the severe phenotype of *TRX m*-silenced plants in an *ntrc* background [40] indicates that *m*-type Trxs are probably subjected to this additional level of regulation. Thus, the functions of the most abundant chloroplast Trxs (types *m* and *f*) but also of the less-abundant type-*x* Trx are affected by NTRC. In this study, we aimed to investigate whether the NTRC-2-Cys Prx system acts as a master regulator of the pool of chloroplast Trxs by extending these analyses to type *y* Trxs. *Arabidopsis* contains two Trx *y* isoforms, *y*1 and *y*2 [41]. These isoforms were unable to activate Calvin–Benson enzymes such as fructose bisphosphatase (FBPase) or sedoheptulose bisphosphatase (SBPase) but were able to reduce thiol peroxidases such as Prx Q and glutathione peroxidase, suggesting an antioxidant function for these Trxs [7,10,27,41,42]. Consistent with this notion, plants deficient in Trxs *y* showed growth phenotypes very similar to the wild type under standard growth conditions [13] but decreased tolerance to high light [13] or drought [43] treatments. To address whether the activity of Trxs *y* is modulated by the NTRC-2-Cys Prx redox system, we evaluated the genetic interaction of Trxs *y* with NTRC through the generation of the *ntrc-trxy1y2* triple mutant. To gain a deeper insight into the functional relationship between these enzymes, this novel mutant was characterized in comparison with the previously reported *ntrc-trxx* mutant [30], as Trxs *x* and *y* are functionally related [32,41,42] and accumulate to a similar range of concentration within chloroplasts [9]. The Trx *y*-deficient mutant, *trxy1y2*, displayed a phenotype similar to that of wild type plants, and the combined deficiencies of NTRC and Trxs *y* resulted in a slight aggravation of the growth retard phenotype of the *ntrc* mutant but did not further affect photosynthesis performance or redox regulation of FBPase or 2-Cys Prxs as compared with the *ntrc* mutant. Furthermore, we performed in vitro and in planta analysis to study the interaction between Trx *y*2, as the most abundant *y* isoform in green tissues, and 2-Cys Prxs. Overall, our results indicate that the functional relationship between Trx *y* and 2-Cys Prxs has subtle effects, if any, on plant physiology.

## 2. Material and Methods

### 2.1. Biological Material, Growth Conditions and Treatments

*Arabidopsis thaliana* wild type (ecotype Columbia) and mutant plants (Appendix A) were grown in soil in growth chambers under long- (16-h light/8-h darkness) or short- (8-h light/16-h darkness) photoperiods at 22 °C and 20 °C during light and dark periods, respectively, and light intensity of 125 µE m^−2^ s^−1^. Mutant plants *trxx, ntrc* and *ntrc-trxx* were reported previously [12,17,30]. The *trxy1y2* mutant was obtained by manually crossing the *trxy1* (SALK_103154) and *trxy2* (SALK_028065) single mutants. To generate the *ntrc-trxy1y2* triple mutant, *ntrc* and *trxy1y2* mutants were manually crossed. Seeds from these crosses were checked for heterozygosity of the T-DNA insertions in the corresponding genes. Plants were then self-crossed, and the double and triple homozygous lines were identified in the progeny by PCR analysis of genomic DNA. Oligonucleotide sequences used for PCR genotyping are listed in Appendix A. For in vitro culture experiments, seeds were surface sterilized using chlorine gas for 16 h, plated on Murashige and Skoog (MS) medium (DUCHEFA), pH 5.8, containing 0.35% Gelrite (DUCHEFA) in the presence or absence of 0.5% (*w*/*v*) sucrose and stratified at 4 °C for 2 to 3 days. *Escherichia coli* and *Agrobacterium tumefaciens* were grown in liquid Miller nutrient at 37 °C and 28 °C, respectively, with the appropriate antibiotics. 

### 2.2. RNA Extraction and RT-PCR Analysis

Total RNA was extracted from rosette leaves of plants grown under long-day conditions for 4 weeks using Trizol reagent (Invitrogen). cDNA synthesis was performed with 1 µg of total RNA using the Maxima first-strand cDNA synthesis kit (Thermo Scientific) according to manufacturer’s instructions. Real-time PCR (RT-PCR) was performed with the GoTaq DNA polymerase (Promega) using primers listed in Appendix A.

### 2.3. Cloning of TRX Y2 and TRX X cDNAs from Arabidopsis. Expression of Recombinant Proteins and Biochemical Assays

For the expression of recombinant proteins, the cDNAs encoding *Arabidopsis* Trx *y*2 and Trx *x*, excluding the predicted transit peptides, were amplified with iProof™ High-Fidelity DNA Polymerase (Bio-Rad) using the oligonucleotides specified in Appendix A, which added a *Bam*HI at the 5′and a *Hind*III site (*TRX Y2*) or *Pst*I (*TRX X*) site at the 3′ ends, respectively. PCR products were gel-purified, cloned in pGEMt vector (Promega) and sequenced. pGEMt-derived plasmids were digested with *Bam*HI and *Pst*I (*TRX Y2*) or *Bam*HI and *Hind*III (*TRX X*), subcloned in pQE30 (Qiagen) expression vector and introduced into *E. coli* XL1Blue. *Arabidopsis* Trx *y*2 and Trx *x* and rice 2-Cys Prx [22] proteins, containing a His-tag at the N-terminus, were purified by NTA affinity chromatography in His-Trap affinity columns (GE Healthcare). For biochemical assays, recombinant 2-Cys Prx (25 μM) was incubated for 30 min with the indicated concentrations of Trxs (*y*2 or *x*) in the presence or absence of 250 μM DTT, as source of reducing power, in reaction buffer (50 mM phosphate buffer, pH 7.4). Free thiols were modified with iodoacetamide (60 mM), and proteins were separated by non-reducing SDS–PAGE using a 14% acrylamide gel and stained with Coomassie blue. 

### 2.4. Bimolecular Fluorescence Complementation (BiFC) Assays

For the generation of BiFC constructs, the full length open reading frame (ORF), excluding the stop codon, encoding Trx *y*2 and FBPase were amplified with iProof™ High-Fidelity DNA Polymerase (Bio-Rad) using oligonucleotides specified in Appendix A, which added attB recombination sites at the 5′ and 3′ ends, respectively. The PCR products were cloned in the Gateway vector pDONR207 (Invitrogen) and sequenced. The cloned Trx *y*2 and FBPase fragments were then transferred to the BiFC vectors [44] pSPYNE-*35S_GW* and pSPYCE-*35S_GW*, respectively, using the LR clonase (Invitrogen). The resulting plasmids, pSPYNE:Trx *y*2 and pSPYCE:FBPase, were then transformed into the *A. tumefaciens* strain GV301. For BiFC assays, *Agrobacterium* strains carrying individual Trx *y*2, FBPase, Trx *x* [45], Trx *f*1 [25], glutamine synthetase 2 (GS2) [28] or 2-Cys Prxs [45] constructs were mixed at a 1:1 ratio and infiltrated into leaves of 4-week-old *Nicotiana benthamiana* plants. Leaf sections were analyzed 4 days later by confocal microscopy performed with a Leica SP/2 inverted microscope. Image analysis was performed with the Leica SP/2 software package and the ImageJ bundle provided by the Wright Cell Imaging facility.

### 2.5. Root Growth Measurements

For root growth assays, three biological replicates of at least 30 seedlings were grown for 7 days on vertically oriented MS plates in the presence or absence of 0.5% (*w*/*v*) sucrose. Root elongation was scored at 1-day intervals from day 3 until day 7. For oxidative stress treatments, sucrose-containing plates were supplemented with 0.025 µM methyl viologen, and root elongation was scored at 12 days. 

### 2.6. Protein Extraction, Alkylation Assays and Western Blot Analysis

Rosette leaves from plants grown under short-day conditions for 6 weeks were ground with a mortar and pestle under liquid nitrogen. Extraction buffer (50 mM Tris-HCl pH 8.0, 0.15 M NaCl, 0.5% (*v*/*v*) Nonidet P-40) was immediately added, mixed on a vortex and centrifuged at 16,100× *g* at 4 °C for 20 min. Protein was quantified using Bradford reagent (Bio-Rad). Rosette leaves from plants grown under long-day for 4 weeks were used for alkylation assays, which were performed as previously described [31] using 60 mM iodoacetamide (FBPase) or 10 mM methylmaleimide-(polyethylene glycol)*_24_* (MM(PEG)_24_) (2-Cys Prxs). Protein samples were subjected to SDS-PAGE under reducing (NTRC, Trx *y*2 and 2-Cys Prxs) or non-reducing (FBPase) conditions using acrylamide gel concentration of 9.5% (FBPase), 14% (2-Cys Prxs) and 15% (NTRC and Trx *y*2). Resolved proteins were transferred to nitrocellulose membranes and probed with the indicated antibody. Specific antibodies for NTRC and 2-Cys Prxs were previously raised in our laboratory [17,22]. The antibody for Trx *y*2 was raised by immunizing rabbits with purified peptides (CQLVERIENSLQVKQ and CQLIQRIEDSLKVKP) by GenScript USA Inc (Piscataway, NJ, USA). The anti-FBPase antibody was kindly provided by Dr. Sahrawy (Estación Experimental del Zaidín, Granada, Spain). 

### 2.7. Determination of Chlorophylls and Measurements of Chlorophyll a Fluorescence

Chlorophyll levels were measured as previously described [22] in plants grown for 4 weeks (long-day) or 6 weeks (short-day). Room temperature chlorophyll a fluorescence was performed using a pulse-amplitude modulation fluorometer (IMAGING-PAM M-Series instrument, Walz, Effeltrich, Germany) on plants grown under short-day for 6 weeks. Induction–recovery curves were performed using blue (450 nm) actinic light (81 μE m^−2^ s^−1^) at the intensities specified for each experiment for 6 min. Saturating pulses of blue light (10,000 μE m^−2^ s^−1^) and 0.6 s duration were applied every 60 s, and recovery in darkness was recorded for another 6 min. The maximum photosystem II (PSII) quantum yield, determined as variable fluorescence (F_v_) to maximal fluorescence (F_m_), F_v_/F_m_, the parameters Y(II) and Y(NPQ), corresponding to the respective quantum yields of PSII photochemistry and non-photochemical quenching, were calculated by ImagingWin v2.46i software according to the equations previously described [46]. Relative linear electron transport rates were measured in leaves of pre-illuminated plants by applying stepwise increasing actinic light intensities up to 1251 μE m^−2^ s^−1^. 

## 3. Results and Discussion

### 3.1. Depletion of Trxs y Aggravates the Growth Inhibition Phenotype of NTRC-Deficient Plants at Adult Stages

Previous genetic studies have established that decreased levels of chloroplastic Trxs, e.g., *x* or *f*, result in subtle phenotypic consequences [12,15]. On the contrary, plants combining the deficiencies of NTRC and Trxs of the *f-* or *x*-type display severe growth inhibition phenotypes [29,30], suggesting that NTRC and plastid Trxs act concertedly. The fact that decreased levels of 2-Cys Prxs partially suppressed the phenotype of NTRC-deficient plants [31,37] led us to propose that NTRC exerts its function on chloroplast redox homeostasis through the maintenance of the reductive capacity of Trxs. However, it is not yet known whether the NTRC-2-Cys Prx redox system controls the reducing capacity of another plastid Trxs. To address this issue, in this work we focused on type *y* Trxs, of which *Arabidopsis* contains two genes, *TRX Y1* and *TRX Y2*, differentially expressed in roots and leaves, respectively [41]. By analyzing the genetic interaction of NTRC with *y*-type Trxs, we pursued two goals: (i) to identify any specific role of Trxs *y*, as the activity of a particular Trx becomes more relevant in the absence of NTRC, and (ii) to test whether *y*-type Trxs are also under the redox regulation of the NTRC-2-Cys Prx system. To address these objectives, we first generated the *trxy1y2* double mutant by crossing of the single *trxy1* and *trxy2* mutants. Next, the *trxy1y2* was crossed with the *ntrc* mutant to obtain the *ntrc-trxy1y2* triple mutant (Figure 1A). Western blot analysis showed the absence of NTRC and Trx *y*2 in the *ntrc-trxy1y2* triple mutant (Figure 1B). As the anti-Trx *y*1 antibody is not available, the absence of *TRX Y1* transcript in the *trxy1y2* and *ntrc-trxy1y2* mutants was confirmed by RT-PCR (Appendix A). Because the phenotype of NTRC-deficient plants is highly influenced by the photoperiod [22,31,47], the vegetative growth of these lines was evaluated both under short-day (Figure 1A,C) and long-day (Appendix A) conditions. Based on the fact that Trx *x* is functionally related to Trxs *y* [32,41,42] and both Trxs accumulate to a similar range of concentration in the chloroplast stroma [9], the previously reported mutants *trxx* [12] and *ntrc-trxx* [30] were included in these analyses for comparison. Plants deficient in Trx *x* or Trxs *y*, *trxx* and *trxy1y2* mutants, respectively, essentially showed a wild type phenotype under either photoperiod in terms of fresh weight (Figure 1A,C; Appendix A), which confirms earlier observations [12,13,30]. As expected, plants lacking NTRC displayed its characteristic, well-known phenotype consisting of retarded growth and pale green leaves, which was more severe under short-day photoperiod (Figure 1B,C). Interestingly, the combined deficiency of NTRC and Trxs *y* led to a slight, but statistically significant, decrease in growth rate compared to *ntrc* plants, which was more remarkable under short-day (Figure 1C) than in long-day (Appendix A) conditions. However, the growth inhibition phenotype of the *ntrc-trxy1y2* mutant was less-severe than that of the *ntrc-trxx* mutant (Figure 1A,C; Appendix A). Both NTRC and Trx *y* have been shown to reduce CHLI [25,27], a component of the Mg-chelatase complex involved in chlorophyll biosynthesis. Hence, it could be expected that accumulation of photosynthetic pigments may be further compromised by depletion of these enzymes. On the contrary, chlorophyll levels in *ntrc-trxy1y2* plants remained similar to those observed in the *ntrc* mutant (Figure 1D; Appendix A), suggesting no contribution of Trxs *y* to chlorophyll biosynthesis. Altogether, these results indicate that the absence of Trxs *y* slightly aggravates growth inhibition in the *ntrc*, but not in the wild-type background, thus lending support to the notion that NTRC may affect the function of these enzymes.

It was previously reported that lines simultaneously devoid of NTRC and Trxs, *x* or *f,* showed compromised viability at the seedling stage [30], uncovering the important role of chloroplast redox regulation during early plant development. Photosynthetic activity of cotyledon chloroplasts provides sucrose, allowing seedling establishment and the formation of vegetative organs, such as roots [48]. Thus, to test the relevance of Trxs *y* at early stages of development, we monitored the formation of roots, as sink organs, in seedlings grown in vitro as a readout of redox homeostasis within plastids. In the absence of an exogenous carbon source, seedlings of the *trxx* and *trxy1y2* mutants formed roots, which showed growth rates similar to that of the wild type (Figure 2A). Notably, root elongation was similarly impaired in *ntrc* and *ntrc-trxy1y2* seedlings, whereas the *ntrc-trxx* mutant barely produced roots (Figure 2A). In agreement with previous reports [30,37], the addition of sucrose increased the rate of root growth in all lines analyzed (Figure 2B), supporting the relevant contribution of photosynthetic activity in cotyledons to the formation of sink tissues [48]. In particular, the rate of root growth in sucrose-containing plates resembled that observed in the absence of the exogenous carbon source, root length being similar in wild type and *trxy1y2*, slightly decreased in *trxx* and *ntrc* and, to a greater extent, in *ntrc-trxy1y2* and *ntrc-trxx* mutants (Figure 2B). Therefore, in plants lacking NTRC, the additional absence of Trx *x* provoked a clear arrest of seedling growth; in contrast, the additional depletion of Trxs *y* had no remarkable effect, revealing that the activity of Trxs *y* is not critical during early stages of plant development.

### 3.2. Trxs y and x Have Differential Effects on Photosynthetic Performance and Redox Regulation of FBPase under Standard Growth Conditions

Once established that in plants lacking NTRC the additional deficiency of Trxs *y* causes an effect at the adult, but not at the seedling stage, we searched for the molecular basis of this effect. Previous work uncovered the effect of chloroplast redox homeostasis on the regulation of light-driven photochemistry reactions, remarkably impaired in mutants lacking NTRC but not in mutants deficient in individual Trxs [29,49]. Moreover, further analysis showed that these phenotypes were severely aggravated in mutant plants combining the deficiencies of NTRC and *x*- or *f*-type Trxs [30]. To assess whether the activity of Trxs *y* affects photosynthetic performance, different chlorophyll fluorescence parameters were determined in plants grown under short-day, in which the growth of the *ntrc-trxy1y2* mutant is impaired to a greater extent (Figure 1A,C). First, we determined the quantum yields of PSII photochemistry (Y(II)) and non-photochemical quenching (Y(NPQ)) during photosynthetic induction. Light energy utilization in the *trxy1y2* mutant was comparable to that observed in the wild type, whereas the *ntrc-trxy1y2* mutant showed Y(II) (Figure 3A) and Y(NPQ) (Figure 3B) values similar to those in the *ntrc* mutant, indicating no effect of Trxs *y* on photosynthetic performance under the tested conditions. Next, we determined the light response curve of electron transport rate (ETR) through the photosystem (PS) II at increasing light intensities (Appendix A). In line with their growth phenotypes, ETR was barely affected in mutant plants devoid of Trxs *y*, which showed values comparable to those in the wild type, though a slight decrease was observed at higher light intensities, in contrast with the *trxx* mutant, which showed a slight decrease of ETR at any light intensity (Appendix A). ETR was markedly lowered and abolished at higher light intensities in the NTRC-deficient mutants, *ntrc* and *ntrc-trxy1y2*, which showed indistinguishable values of ETR, and the *ntrc-trxx* mutant, which displayed a more severe effect (Appendix A). These results indicate that, regardless of light intensity, Trxs *y* have a minor contribution to the rate of electron transport at least under the tested conditions. Finally, we determined the F_v_/F_m_ ratio, as indicative of the stability of the PSII (Table 1). Mutant plants lacking Trxs *x* or *y* showed F_v_/F_m_ values that were very similar to those observed in the wild type. On the contrary, F_v_/F_m_ was remarkably decreased in the *ntrc* mutant, the additional deficiency of Trxs *y* causing a slight decrease, which is in contrast with the higher effect caused by the additional absence of Trx *x*. Taken together, chlorophyll fluorescence measurements indicate that Trxs *y* exert a minor contribution to the photosynthetic performance of plants. Additionally, our results show that in the *ntrc* mutant background, the additional deficiency of Trxs *x*, but not of Trxs *y*, aggravated photosynthetic parameters, which is in line with the growth phenotypes of the *ntrc-trxx* and *ntrc-trxy1y2* mutants. 

Plant growth requires the adaptation of chloroplast metabolism to light availability, which is based in light-activation of thiol-modulated enzymes, such as FBPase [50]. Accordingly, impaired redox regulation of this enzyme correlates with a marked limitation of growth [29,30]. The fact that depletion of Trxs *y* slightly aggravates the growth inhibition phenotype of NTRC-deficient plants (Figure 1A,C and Appendix A) without affecting their photosynthetic performance (Figure 3A,B, Appendix A and Table 1), raises the possibility that the activity of Trxs *y* affected redox regulation of metabolic pathways. To address this issue, the redox balance of FBPase, a key regulatory enzyme of the Calvin–Benson cycle, was analyzed during transitions from darkness to light intensity of 150 µE m^−2^ s^−1^ (growth light) or 450 µE m^−2^ s^−1^ (high light). Thiol-labelling assays using iodoacetamide (IAA) showed that FBPase, which was completely oxidized in darkness, became partially reduced under growth-light in both wild-type and *trxy1y2* plants (Figure 4A,B). On the contrary, mutant plants devoid of NTRC or, to a lesser extent, Trx *x* showed impaired light activation of FBPase. Under high-light intensity, the level of reduced enzyme increased, FBPase being almost fully reduced in the wild-type and the *trxy1y2* mutant. Likewise, light-dependent FBPase reduction was recovered in *ntrc-trxy1y2* and *ntrc* mutants and, to a greater extent, in the *trxx* mutant. In agreement with previous studies [30,37], FBPase reduction was almost abolished in the *ntrc-trxx* mutant, regardless of the light intensity (Figure 4A,B). Overall, the degree of reduction of FBPase correlated with the photosynthetic electron transport rates (Appendix A). 

The fact that depletion of Trxs *y* barely affected the level of reduction of FBPase is in line with biochemical analysis showing that type *y* Trxs are poor reductants of FBPase [10,41]. Likewise, Trx *x* is an inefficient reductant of FBPase in vitro [10,30,32], and, accordingly, these enzymes do not interact in planta [26]. Therefore, both Trx *x* and Trxs *y*, similarly abundant in the chloroplast stroma, are inefficient reductants of FBPase, yet the absence of Trx *x*, but not of Trxs *y*, causes impairment of light dependent reduction of the enzyme. However, while the impairment of FBPase reduction observed in the *ntrc* mutant was severely aggravated in the *ntrc-trxx* mutant, no additional effect was observed in the *ntrc-trxy1y2* mutant (Figure 4A,B), indicating different effects of Trx *x* and Trxs *y* on FBPase redox regulation. We have recently proposed that the absence of NTRC affects light-dependent FBPase reduction due to drainage of reducing equivalents from the pool of Trxs via reduction of 2-Cys Prxs, which thus affects regulation of downstream targets [31]. Our results suggest that the absence of Trx *x* would cause higher drainage of reducing equivalents from Trxs *f* and *m*, than the absence of Trxs *y*, thereby causing higher impairment of FBPase reduction. To explore this possibility, we analyzed the interaction of Trxs *y* with 2-Cys Prxs.

### 3.3. Trx y2 Interacts with 2-Cys Prxs In Vitro and In Planta

Extensive biochemical analyses have shown that most plastidial Trxs have the ability to reduce 2-Cys Prxs [22,27,32,33,34,35,36]. Nevertheless, the in vivo contribution of these enzymes to maintain the redox balance of 2-Cys Prxs remains unknown. Thus, to gain more insight into this issue, we explored the interaction between Trxs *y* and 2-Cys Prxs. Similar to other Trxs, Trx *y*1 was shown to reduce 2-Cys Prx in vitro [27,41]; however, the *TRX Y1* gene is mainly expressed in non-photosynthetic organs [41], and, accordingly, the Trx *y*1 protein was not detected in the chloroplast stroma of *Arabidopsis* leaves [9]. Therefore, Trx *y*2, which shows predominant expression in leaves [41], was chosen for subsequent experiments. We first performed bimolecular fluorescence complementation (BiFC) experiments in *Nicotiana benthamiana* leaves to test the interaction of Trxs *y* and 2-Cys Prxs in planta. To that end, we generated constructs to express Trx *y*2 fused to the N-terminal part of YFP (Trx *y*2-_N_YFP) and took advantage of available plasmids [45] expressing Trx *x*-_N_YFP, here included as a positive control, and 2-Cys Prx A and B fused to the C-terminal part of YFP (2-Cys Prx A-_C_YFP and 2-Cys Prx B-_C_YFP). Co-infiltration of Trx *x*-_N_YFP with either 2-Cys Prx A-_C_YFP or 2-Cys Prx B-_C_YFP constructs resulted in a chloroplast-localized yellow fluorescence in *Nicotiana* leaves (Appendix A), confirming previous results [45]. Notably, reconstitution of the fluorescent signal within chloroplasts was also observed when BiFC vectors expressing Trx *y*2-_N_YFP with either 2-Cys Prx A-_C_YFP or 2-Cys Prx B-_C_YFP were co-infiltrated (Figure 5A), though the YFP signal was weaker than that of the Trx *x* with 2-Cys Prxs (Appendix A). Remarkably, the pattern of interaction of both Trx *y*2 and Trx *x* with 2-Cys PRXs showed the formation of speckles, which may reflect the tendency of 2-Cys Prxs to form aggregates [51]. Additionally, a set of BIFC assays were performed as negative controls to validate the interaction between Trx *y*2 and 2-Cys Prxs. To that end, we made constructs to express FBPase fused to the C-terminal part of YFP (FBPase-_C_YFP) and used plasmids [28] expressing the chloroplast glutamine synthetase 2 (GS2) fused to the N-terminal part of YFP (GS2-_N_YFP). 

Previous work showed that *y*-type Trxs are not involved in the reductive activation of FBPase [10,41], which is efficiently catalyzed by Trxs *f* [27]. In line with these biochemical studies, mutants lacking Trxs *y* showed WT levels of FBPase reduction (Figure 4A,B). Expectedly, background fluorescent signals were detected when Trx *y*2-_N_YFP and FBPase-_C_YFP constructs were infiltrated, whereas reconstitution of an intense YFP signal was observed when Trx *f*1-_N_YFP, used as a positive control, was co-infiltrated with FBPase-_C_YFP (Appendix A). Previous proteomic studies identified putative interacting partners of 2-Cys Prxs [52] and NTRC [28]. Among the interacting candidates, GS2 was identified as a partner of NTRC, but not of 2-Cys Prx [28,52] and thus was selected as a negative control for BIFC assays. In agreement with these findings, no YFP signal was observed when GS2-_N_YFP and 2-Cys Prx-_C_YFP constructs were infiltrated (Appendix A). On the contrary, NTRC did interact with GS2 (Appendix A), confirming previous BiFC analyses [28]. Taken together, these results indicate that, similar to Trx *x* [45], Trx *y*2 interacts with 2-Cys Prxs in planta. 

Next, we aimed to study the redox interaction between Trxs *y* and 2-Cys Prxs by testing the ability of Trx *y2* to reduce 2-Cys Prx in vitro. The recombinant proteins Trx *y*2 and Trx *x,* included here for comparative purposes, and 2-Cys Prx were produced in *E. coli* in its mature form. Non-reducing SDS–PAGE electrophoresis showed that 2-Cys Prx was mainly detected in its oxidized form, a dimer where the two subunits are bound by intermolecular disulphide bond(s) (Figure 5B). Notably, incubation with either Trx, *x* or *y*2, in the presence of a low concentration of DTT, which slightly reduced 2-Cys Prx (Figure 5B), resulted in the reduction of 2-Cys Prx, as the dimeric form of the enzyme shifted to the monomeric form in a concentration-dependent manner, indicating the redox interaction between both enzymes. However, while almost 40% of reduction of 2-Cys Prx was achieved with 0.5 μM concentration of Trx *x*, the concentration of Trx *y*2 to obtain a similar level of 2-Cys Prx reduction was three-fold higher (Figure 5C), indicating that Trx *y*2 reduces 2-Cys Prx with lower efficiency than Trx *x*. 

The fact that Trx *y*2 reduces 2-Cys Prx in vitro (Figure 5B,C) and interacts with both isoforms, A and B, in planta (Figure 5A) raises the question of the functional relationship between these enzymes. To address this issue, we performed thiol-labelling experiments with the alkylation agent methylmaleimide-(polyethylene glycol)*_24_* (MM(PEG)_24_) to monitor the redox balance of 2-Cys Prxs in wild type and mutant plants exposed to different light intensities (Figure 6A). The almost identical *Arabidopsis* 2-Cys Prxs, A and B, arrange as head-to-tail homodimers, each monomer containing two Cys residues, peroxidatic and resolving, that participate in catalysis [53]. Therefore, in alkylation experiments, 2-Cys Prxs are visualized in three different bands, the upper and intermediate bands corresponding to the fully reduced and single-disulphide forms, monomers containing two (2 SH) or one (1 SH) Cys in its thiolic form, respectively. The lowest, unshifted, band corresponds to the double-disulphide form (0 SH), hence the fully oxidized enzyme. Dark-to-light transitions led to slightly increased levels of the reduced forms of 2-Cys Prxs; however, the redox state of these enzymes, regardless of the light condition, remained similar in all the lines analyzed, including the *ntrc* mutant (Figure 6A,B). 

These observations agree with previous results [37,54] showing that the deficiency of NTRC resulted in minor differences in the redox state of 2-Cys Prx in either dark or light conditions. It should be emphasized, however, that it was previously reported that 2-Cys Prxs were remarkably oxidized in dark-adapted leaves of *ntrc* mutants grown under a short-day photoperiod [26,31]. These discrepancies might be explained by differences in plant material (whole rosettes or leaves) and/or growth conditions (long- or short-day photoperiod). In this regard, it was recently reported that photosynthetic parameters, and probably the content and/or redox state of chloroplast proteins, are impaired to different extents in young and mature leaves of *ntrc* mutants [54]; thus leaf age may also contribute to the variability of the 2-Cys Prx redox state determinations. Nevertheless, our results support the notion that 2-Cys Prxs act as a sink of reducing equivalents from Trxs, which become more prominent in the absence of NTRC. Indeed, the *ntrc-trxx* mutant, which is simultaneously deficient in NTRC and Trx *x*, exhibited decreased levels of the reduced forms of 2-Cys Prxs, supporting the notion that these enzymes are efficiently reduced by NTRC and Trx *x*. On the contrary, 2-Cys Prx reduction was similarly affected in *ntrc* and *ntrc-trxy1y2* (Figure 6A,B). Thus, the imbalance of the 2-Cys Prxs redox state, slightly more oxidized in the *ntrc-trxx* than in the *ntrc-trxy1y2* mutant, is expected to increase the drainage of reducing equivalents from Trxs *m* and *f*, thereby impairing the light dependent reduction of FBPase in the *ntrc-trxx* mutant, but not in the *ntrc-trxy1y2* mutant (Figure 4A,B). Taken together, these combined results suggest that the activity of Trxs *y* exert a minor contribution, if any, on the redox state of 2-Cys Prxs. 

### 3.4. The Absence of Trxs y Barely Affects the Response of Plants to Oxidative Stress

The results described above indicate that the activity of Trxs *y*, regardless of the photoperiod and developmental stage, exerts a moderate contribution to plant performance under standard growth conditions. Nevertheless, earlier work proposed the participation of Trxs *y* in the antioxidant response under several abiotic stresses. In particular, the activity of the methionine sulfoxide reductase, an enzyme mediating the regeneration of the reduced form of methionine in proteins, was lowered in mutant plants devoid of Trxs *y* grown under high light [13]. Moreover, depletion of Trxs *y* led to an increased oxidation of the ascorbate pool under water deprivation [43]. Hence, we aimed to examine whether NTRC affects the contribution of Trxs *y* to chloroplast protection against oxidative stress. To that end, we evaluated the effect of methyl viologen (MV), which induces the production of ROS within illuminated chloroplasts under different plant developmental stages. The oxidative stress response during early stages was tested by monitoring root formation of seedlings grown in plates supplemented with methyl viologen (Appendix A). While root growth in the presence of the photooxidative agent was impaired in all lines under study, both *ntrc* and *ntrc-trxy1y2* mutants displayed a similar response to the treatment as shown by the rate of root growth of seedlings. Finally, the oxidative response at the adult stage was examined by monitoring the F_v_/F_m,_ defining the maximal PSII quantum yield, in MV-sprayed plants (Figure 7A,B). The general trend was that after 1 day of the treatment, all lines under analysis responded similarly as shown by decreased F_v_/F_m_ values. Interestingly, after 4 days of the treatment, F_v_/F_m_ was partially recovered in wild type, and, to a lower extent, in *trxx* and *trxy1y2* mutants, but was further decreased in single and multiple *ntrc* mutants. The response to MV depends on ROS-scavenging enzymes, such as chloroplast peroxidases [55]. Biochemical studies reported that several chloroplast enzymes with antioxidant function receive reducing power from type *y* Trxs. In particular, both Trx *y* isoforms are capable of reducing Prx Q [7,10,27,41] and glutathione peroxidase [42], suggesting that Trxs *y* may function as in vivo reductants for these enzymes. Nevertheless, the fact that the oxidative stress response is unaffected by depletion of Trxs, *x* or *y,* and similarly impaired in the *ntrc*, *ntrc-trxx* and *ntrc-trxy1y2* mutants, suggests the predominant role of NTRC in abiotic stress tolerance. 

## 4. Conclusions

Previous analyses of the genetic interaction of NTRC and chloroplast Trxs have shown that simultaneous deficiencies of these enzymes result in a dramatic inhibition of growth [29,30,40], which correlates with the disruption of the redox homeostasis in the organelle [31,37]. One conclusion drawn from these studies was that the activity of NTRC is required for the function of different Trxs, namely the *f*, *x* and, probably, the *m* subtypes. In this study, we explored whether NTRC acts as a master regulator of the pool of chloroplast Trxs by analyzing its functional relationship with Trxs *y*. The current knowledge of redox regulation of photosynthetic metabolism (recently reviewed in [24]) establishes that the NTRC-2-Cys Prx system controls the reducing capacity of chloroplast Trxs and, consequently, the reductive activation of their target enzymes during the day [26,31,37] and its oxidative inactivation during the night [36,38,39]. Trx *y*2 is able to transfer reducing equivalents to 2-Cys Prxs and interact with them in planta; thus, a possible explanation for the decreased growth rosettes in the *ntrc-trxy1y2* mutant could be a further redox imbalance of 2-Cys Prxs in these plants, which potentially would impair regulation of a large set of chloroplast enzymes. If this was the case, the combined deficiencies of NTRC and Trxs *y* would aggravate the redox imbalance of 2-Cys Prxs and, consequently, of downstream targets such as FBPase. Nevertheless, the in vivo redox state of neither 2-Cys Prxs nor FBPase was significantly affected in the absence of Trxs *y*, questioning the physiological relevance of the interaction between Trxs *y* and 2-Cys Prxs. Finally, we cannot rule out the possibility that these phenotypic effects are caused by selectivity toward a particular chloroplast enzyme(s). In this regard, Trxs *y* can transfer reducing equivalents to additional thiol peroxidases such as Prx Q [7,10,27,41] and glutathione peroxidases [42], among other targets [7,13,43]. The physiological significance of these activities should be investigated in future studies.

## Figures and Tables

**Figure 1 antioxidants-09-01072-f001:**
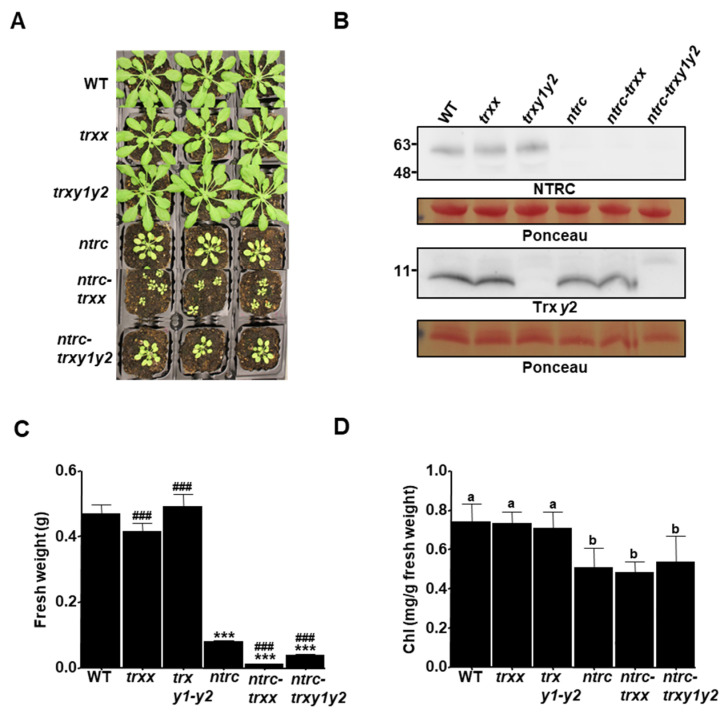
Growth phenotypes of mutant plants devoid of Trxs *y*. (**A**) Wild-type (WT) and mutant lines, as indicated, were grown under short-day conditions for 6 weeks. (**B**) Western blot analysis of the content of NTRC and Trx *y*2. Protein extracts obtained from leaves of plants grown as stated in (**A**), were subjected to SDS–PAGE under reducing conditions, transferred to nitrocellulose filters and probed with anti-NTRC or anti-Trx *y*2 antibodies. Even loading was monitored by Ponceau staining of the Rubisco large subunit and molecular weight markers (kDa) are indicated on the left. The fresh weight (**C**) and the content of chlorophylls (**D**) of rosette leaves from fourteen plants of each line are represented as average values ± standard error (SE). Statistical significance compared with the wild type (asterisks) or the *ntrc* mutant (hashes) is indicated (*** and ###, *P <* 0.001, Student’s *t* test). Letters indicate significant differences (*P* < 0.05) determined by one-way ANOVA followed by Tukey’s post-test.

**Figure 2 antioxidants-09-01072-f002:**
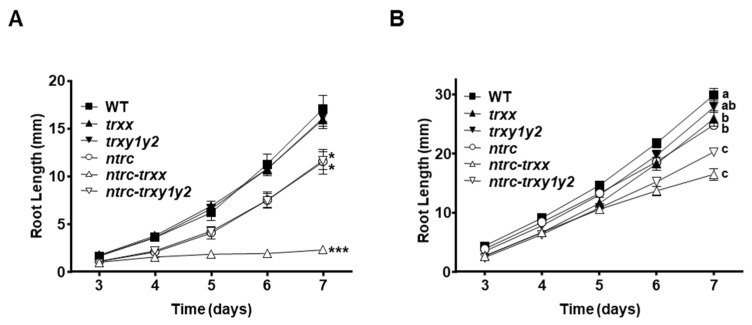
The contribution of Trxs *y* to seedling root formation. Seedlings of wild-type (WT) and mutant lines, as indicated, were grown under long-day photoperiod on Murashige and Skoog (MS) synthetic medium in the absence (**A**) or presence (**B**) of 0.5% (*w*/*v*) sucrose. Root growth was monitored at the indicated days. Mean values of three independent replicates ± standard error (SE) are represented. Statistical significance compared with the wild type is indicated by asterisks (*, *P* < 0.05 and ***, *P* < 0.001, Student’s *t* test). Letters indicate significant differences (*P* < 0.05) determined by one-way ANOVA followed by Tukey’s post-test.

**Figure 3 antioxidants-09-01072-f003:**
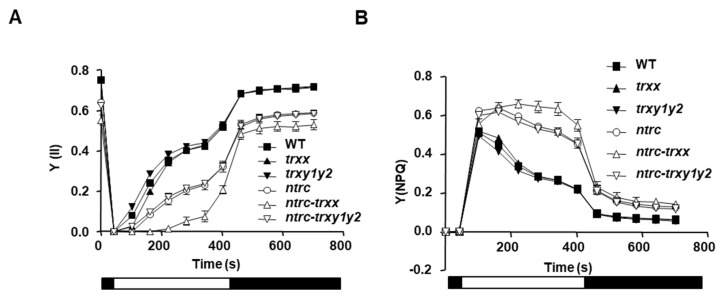
Effect of Trxs *y* on photosynthetic performance. Photosynthetic parameters were measured on whole rosettes of the wild type and mutant lines, as indicated, grown under short-day conditions for 6 weeks. (**A**) Quantum yields of photosystem II photochemistry (Y(II)) and (**B**) non-photochemical quenching (Y(NPQ)). Each value is the average of three determinations, except for *ntrc-trxx* (four measurements). Standard errors of the mean (SEM) are represented as error bars. White and black blocks indicate periods of illumination with actinic light (81 μE m^−2^ s^−1^) and darkness, respectively.

**Figure 4 antioxidants-09-01072-f004:**
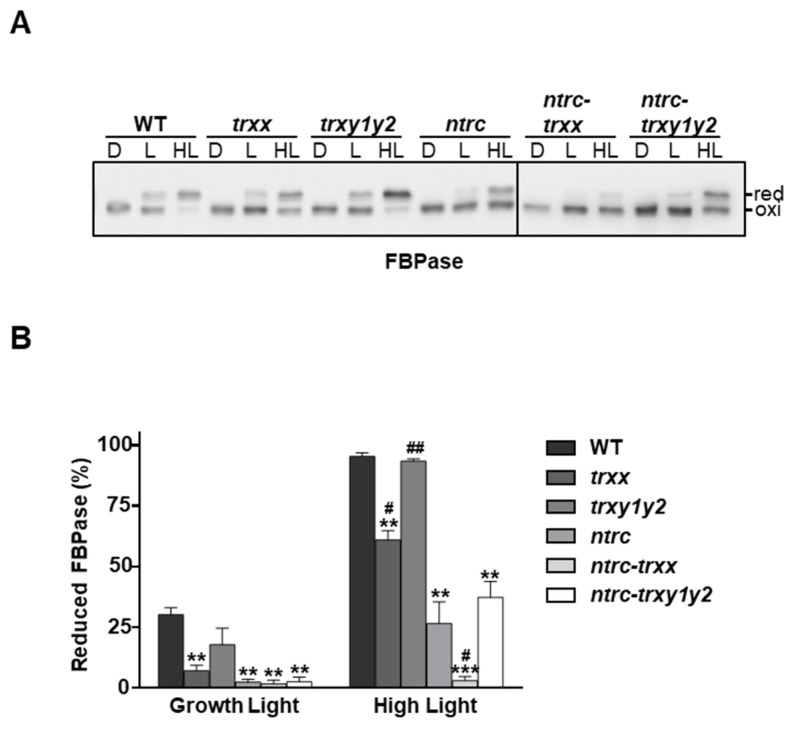
In vivo redox state of FBPase in mutant plants devoid of Trxs *y*. Plants of the wild type and mutant lines, as indicated, were grown under long day conditions for 4 weeks. Whole rosettes were harvested at the end of the period of darkness (D) and then illuminated for 30 min with light intensities of 125 μE m^−2^ s^−1^ (growth light, L) or 450 μE m^−2^ s^−1^ (high light, HL). (**A**) The in vivo redox state of FBPase was determined by alkylation using the thiol-labelling agent iodoacetamide (IAA). Protein extracts were subjected to SDS–PAGE under non-reducing conditions, transferred onto nitrocellulose filter and probed with an anti-FBPase antibody. (**B**) Band intensities were quantified (GelAnalyzer), and the percentage of reduction was calculated as the ratio between the reduced form and the sum of oxidized and reduced forms. Each value is the mean of three independent experiments ± standard error (SE). Statistical significance compared with the wild type (asterisks) or the *ntrc* mutant (hashes) is indicated (#, *p* < 0.05; ** and ##, *p* < 0.01; ***, *p* < 0.001, Student’s *t* test); red, reduced; oxi, oxidized.

**Figure 5 antioxidants-09-01072-f005:**
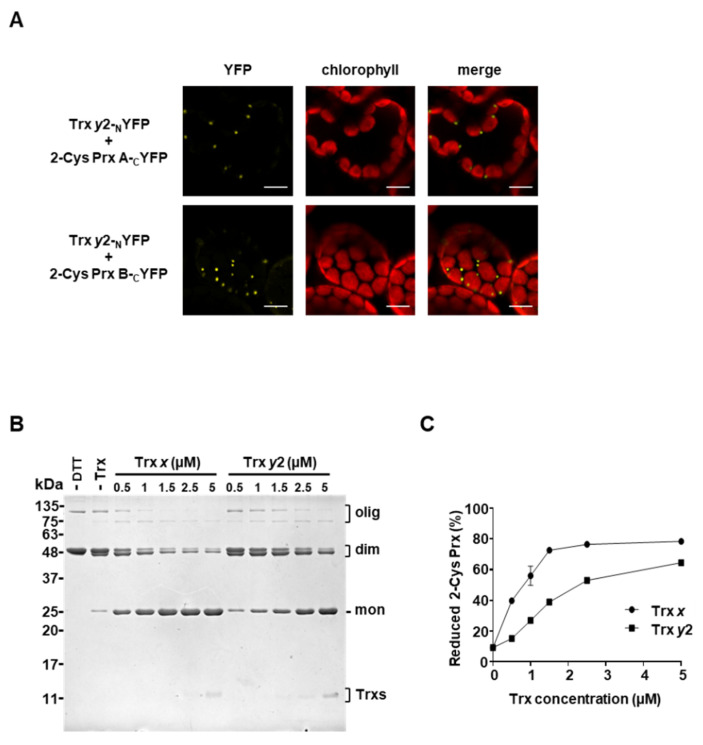
Analysis of the interaction of Trx *y*2 with 2-Cys Prxs in planta and in vitro. (**A**) BiFC analysis of the interaction of Trx *y*2 with the two *Arabidopsis* 2-Cys Prxs isoforms, A and B. Confocal microscopy micrographs of mesophyll cells of *Nicotiana benthamiana* leaves acquired 4 days after agro-infiltration with the indicated constructs. Red, chlorophyll autofluorescence; yellow, YFP fluorescence. Bars correspond to 10 μm. (**B**) Redox interaction between Trx *y*2 and 2-Cys Prx. Recombinant 2-Cys Prx (25 μM) from rice was incubated, in the presence or absence of DTT (250 Μm), with the indicated concentrations (0–5 μM) of Trx *y*2 or Trx *x* for 30 min. Thiols were blocked with iodoacetamide, and samples were then subjected to non-reducing SDS–PAGE and stained with Coomassie blue. The reduction of 2-Cys Prx is visualized as the shift from the dimeric (dim) to the monomeric (mon) form of the enzyme, corresponding to the oxidized and reduced forms, respectively. Molecular weight markers (kDa) are indicated on the left; mon, monomer; dim, dimer; olig, oligomer. (**C**) Band intensities were quantified (GelAnalyzer) and the percentage of the reduction level of 2-Cys Prx was calculated as the ratio between the monomeric form and the total. Each value is the mean of three independent experiments ± standard error (SE).

**Figure 6 antioxidants-09-01072-f006:**
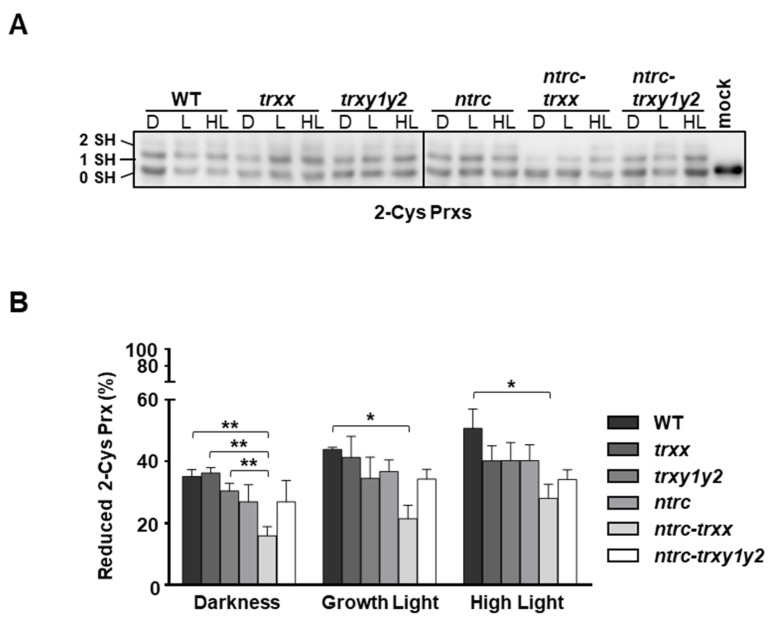
In vivo redox state of 2-Cys Prxs in mutant plants devoid of Trxs ***y***. Plants of the wild type and mutant lines, as indicated, were grown under long day conditions for 4 weeks. Whole rosettes were harvested at the end of the period of darkness (D) and then illuminated for 30 min with light intensities of 125 μE m^−2^ s^−1^ (growth light) or 450 μE m^−2^ s^−1^ (high light). (**A**) The in vivo redox state of 2-Cys Prxs was determined by alkylation using the thiol-labelling agent methylmaleimide-(polyethylene glycol)_24_ (MM(PEG)_24_). Mock indicates a control sample treated with alkylation buffer without MM(PEG)_24_. Protein extracts were subjected to SDS–PAGE under reducing conditions, transferred onto nitrocellulose filter and probed with an anti-2-Cys Prx antibody; 0 SH, 1 SH and 2 SH indicate reduction of none, one or the two cysteine residues, respectively, of 2-Cys Prxs. (**B**) Band intensities were quantified (GelAnalyzer), and the proportion of reduced protein was calculated as the ratio between the sums of half-reduced and fully reduced forms (1 SH + 2 SH) and the sum of oxidized (0 SH) and reduced forms (1 SH+2 SH). Each value is the mean of three independent experiments ± standard error (SE). Asterisks indicate significant differences between mean values (*, *p* < 0.05 and **, *p* < 0.01, Student’s *t* test).

**Figure 7 antioxidants-09-01072-f007:**
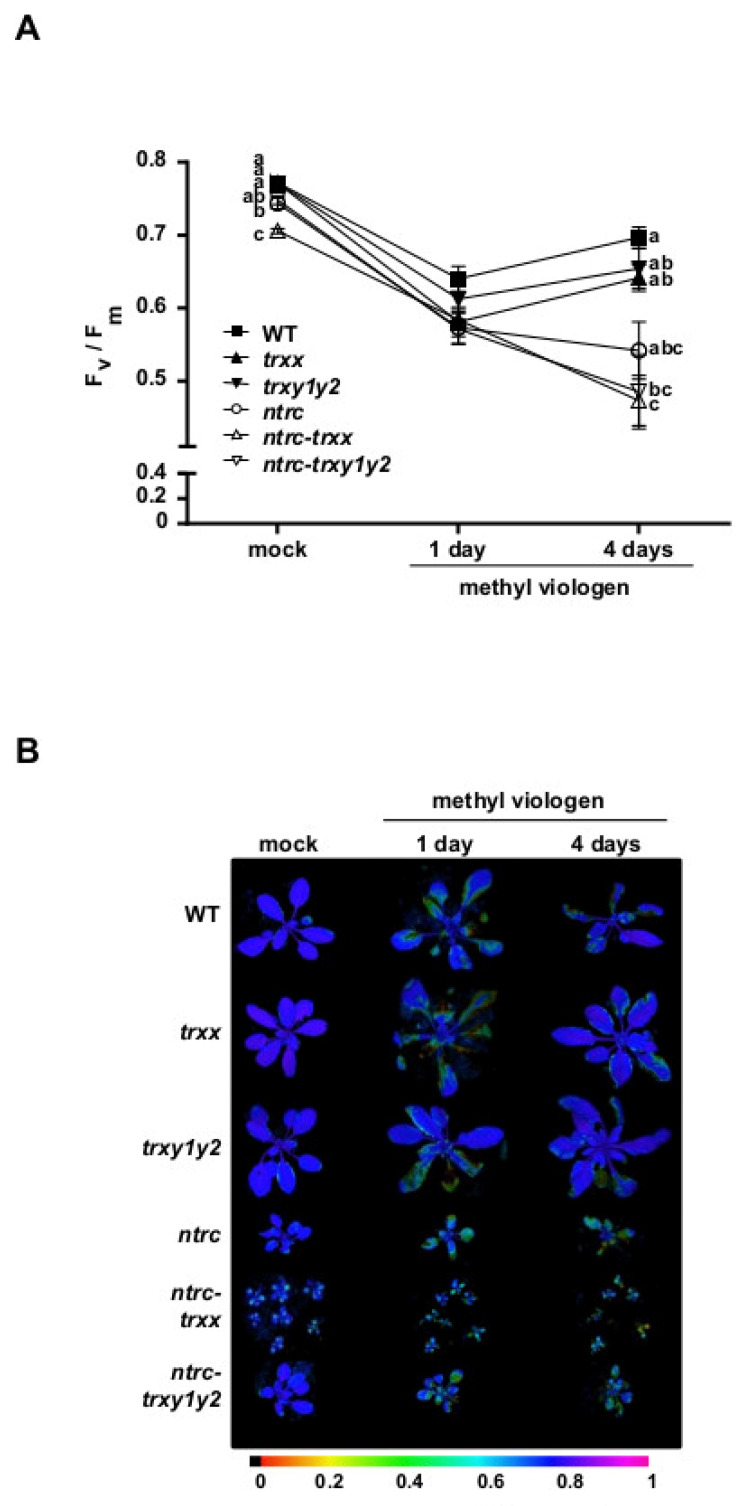
Oxidative stress response in mutant plants devoid of Trxs ***y***. (**A**) Wild type and mutant lines were grown under long-day photoperiod for 4 weeks and sprayed with methyl viologen (15 μM) in 0.025% Tween-20. The F_v_/F_m_, representing the maximum potential quantum efficiency of photosystem II, was measured before (mock) and after (1 and 4 days) the treatment with methyl viologen. F_v_/F_m_ is represented as average values ± standard error (SE) of 7 (mock) or 14 (methyl viologen) plants. Letters indicate significant differences (*p* < 0.05) determined by one-way ANOVA followed by Tukey’s post-test. (**B**) False-color images representing F_v_/F_m_ in mock- and methyl viologen-treated plants. Signal intensities (from 0 to 1.0) are indicated according to the color scale bar.

**Table 1 antioxidants-09-01072-t001:** Effect of the combined deficiencies of NTRC and Trxs *y* on F_v_/F_m_. The maximum PSII quantum yield was determined as variable fluorescence (F_v_) to maximal fluorescence (F_m_), F_v_/F_m_, in dark adapted leaves of plants grown under short-day conditions for 5 weeks. F_v_/F_m_ values (±SD) are the average of measurements from 8 (WT, *trxx* and *trxy1y2*), 15 (*ntrc* and *ntrc-trxy1y2*) and 21 (*ntrc-trxx*) plants. Letters indicate significant differences (*p* < 0.05) determined by one-way ANOVA followed by Tukey’s post-test.

	WT	*trxx*	*trxy1y2*	*ntrc*	*ntrc-trxx*	*ntrc-trxy1y2*
F_v_/F_m_	0.743 ± 0.004 a	0.738 ± 0.006 a	0.718 ± 0.012 ab	0.676 ± 0.005 ab	0.492 ± 0.026 c	0.643 ± 0.011 b

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
