# Peer review of "Exploring the Functional Relationship between y-Type Thioredoxins and 2-Cys Peroxiredoxins in Arabidopsis Chloroplasts"

_antioxidants, 2020, doi:10.3390/antiox9111072_

Round 1

Reviewer 1 Report

The authors have addressed all my previous concerns. I feel that the manuscript now is easy to read and follow and will be of interest to all readers interested in redox biology.

Author Response

I would like to thank the reviewer for the rapid and positive response. 

Reviewer 2 Report

I am happy with the modifications the authors have made for this revised manuscript, with the additional experiments and controls, and with their responses to my previous comments and questions. I also agree that the long light period as well as leaf age very likely had a large effect on 2-CysPrx redox state in all NTRC-deficient lines.

Nevertheless, I still worry that a casual reader (especially a title-and-abstract-only-reader) could come away thinking that the paper argues for a significant role for Trx-y's in maintaining 2CysPrx redox balance. I don't think that a "negative" conclusion makes the contribution of the paper to the research field any less important, and the authors soberly conclude in the discussion that "Taken together, these combined results suggest that the activity of Trxs y exert a minor contribution, if any, on the redox state of 2-Cys Prxs."  In my view, this kind of a statement should also be added to the abstract to make the conclusion clear for busy readers. In the abstract it's written now "this interaction has subtle effects on plant performance", which is decent, but even just changing "subtle" to, for example, "only minor" would make the actual findings more clear already in the abstract.  

Author Response

First, I would like to thank the reviewer for the rapid and positive response. As suggested, the last sentence of the Abstract (lines 27-29) was modified to make clearer that our results indicate that Trxs y do not play a significant role in maintaining the redox balance of 2-Cys Prxs. 

This manuscript is a resubmission of an earlier submission. The following is a list of the peer review reports and author responses from that submission.

Round 1

Reviewer 1 Report

Overall, this is a well written paper with some well conducted experiments and interesting results, but I do think there are several issues that need to be addressed before the manuscript should be considered for publication. First of all, I think that the title of the paper is slightly misleading, and at least in the absence of further evidence should be modified to better reflect the actual findings. As it stands, there is rather little evidence of a functional relationship of any physiological significance between y-type Trxs and 2-CysPrxs. (This is honestly discussed in the Conclusion chapter though). There is in vitro evidence that Trx y’s can reduce 2CysPrxs to some extent (which is previously published), and some evidence in the current manuscript of in vivo interaction of overexpressed Trx y’s and 2CysPrxs in a BiFC system, but it is not possible to directly draw a conclusion of any physiological significance from this. Especially since in my view, better negative control(s) are needed to reliably interpret the obtained results from the BiFC assays. Moreover, the result that the in vivo redox state of FBPase was not any more oxidized in ntrc trxy1y2 in comparison to the ntrc single mutant suggests that removal of Trx y’s likely has very little effect on the redox balance of 2-CysPrxs in vivo (assuming that the oxidizing effect on FBPase is caused via impaired reduction of 2-CysPrxs in these mutants). In any case, the authors should carry out an experiment to determine the in vivo redox state of 2-CysPrxs to clarify this issue.  

Specific points:

Abstract

Line 20-21: Should say decreased 2-CysPrx amount partially rescues the phenotypes. Growth of those plants is still somewhat retarded in comparison to WT, as seen in a previous article from the author’s laboratory (ref [82]). It does seem to recover growth to the level of the 2cp mutant though.

Also,please summarize the key findings of the study in addition to explaining the background and research question in the abstract.

Materials and methods

Lines 170–171: The Y(NPQ) parameter cannot be described as “non-regulated basal quenching”, that would better describe the Y(NO) parameter. Y(NPQ) is the yield of non-photochemical quenching (as correctly used in the Results and discussion section), which is very much a regulated, and a highly dynamic rather than basal process of thermal dissipation of excitation energy.

Please add information about the age of plants used for protein alkylation assays and immunoblots, chlorophyll level determination, and Chl a fluorescence measurements also here in addition to the Figure legends.

Results and Discussion

Lines 181-182: “While the direct interaction of NTRC with chloroplast redox-regulated enzymes cannot be ruled out” – the finding in ref. [26] that overexpression NTRC with an inactivated TRX domain still partially rescues the ntrc phenotype (without increasing the amount of reduced 2-CP) argues strongly in favour of this possibility. Also, I do not think that these two models for the influence pathway of NTRC are mutually exclusive. Both can be true at the same time in vivo.

Figure 3 and Lines 273-274: It should be mentioned in the text too that Y(II) and Y(NPQ) were measured during photosynthetic induction, and not in a light curve like ETR. Also, what is the rational was using ETR in (A) and Y(II) in (B)? ETR is calculated from Y(II) by multiplying with PAR and the fraction of absorbed light and the ratio of PSII/PSI antennae, and I understand that the latter two multipliers were assumed constant between lines? (which is usually done even though there are often major differences in the ratio of PSII/PSI antenna sizes between mutant lines; this is especially true in mutants lacking NTRC, which have a significantly lowered amount of functional PSI: see PsaB protein level in ref. [29] and EPR and 77K spectra in Figures 3 & 4 in Nikkanen et al., 2019 Physiol Plantarum 166: 211-225). Therefore ETR offers really no relevant extra information compared to Y(II) here, while raising questions about the reliability of the values in the mutant lines.  On top of this, In fact, using Y(II) also in (A) would make it possible to better distinguish between differences in the low light intensities, as the PAR value (which are of course same for all lines) in removed from the calculation.

Table I: Why is Fv/Fm so low in WT? Healthy Col-0 plants should very reliably have Fv/Fm of c.a. 0.82-0.84. Where the plants stressed somehow? Also, are the t-tests of significance done in comparison to WT for all mutants? For the double mutants the relevant reference would in my view be the ntrc single mutant.

Lines 284-285: ”..the additional deficiency of Trxs x and y impact photosynthetic parameters to a different extent”- although I suppose technically true, this is slightly misleading, because according to the data shown, additional deficiency of Trxs y in ntrc background has NO extra impact on photosynthetic parameters at all, while in ntrc trxx there is clear aggravation of the phenotype in comparison to ntrc.

Line 343: There should be proper negative controls in the BiFC tests. Lack of interaction between Trx y2 and FBPase would work as a negative control for the Trx y2 side of the positive result, but it is not mentioned as such and does in fact seem to exhibit some YFP fluorescence. The authors should also demonstrate that the 2-CP constructs don’t interact with just everything. This is especially important since the YFP signal from Trx y2 + 2-CysPrxA in particular looks rather weak. In fact, it doesn’t look any stronger than the Trx y2 + FBPase test in Fig S3B, which is interpreted as a negative result.

Conclusion

Line 449; Why was a mobility shift assay not done to examine the in vivo redox state of 2-CysPrxs in new mutant strains, ntrc trxy1y2 in particular? (similarly to the FBPase assay in Fig. 4). This seems central to conclusions of this paper, and if understand correctly it should be no problem to run this experiment in the authors’ laboratory. 

Overall, this is a well written paper with some well conducted experiments and interesting results, but I do think there are several issues that need to be addressed before the manuscript should be considered for publication. First of all, I think that the title of the paper is slightly misleading, and at least in the absence of further evidence should be modified to better reflect the actual findings. As it stands, there is rather little evidence of a functional relationship of any physiological significance between y-type Trxs and 2-CysPrxs. (This is honestly discussed in the Conclusion chapter though). There is in vitro evidence that Trx y’s can reduce 2CysPrxs to some extent (which is previously published), and some evidence in the current manuscript of in vivo interaction of overexpressed Trx y’s and 2CysPrxs in a BiFC system, but it is not possible to directly draw a conclusion of any physiological significance from this. Especially since in my view, better negative control(s) are needed to reliably interpret the obtained results from the BiFC assays. Moreover, the result that the in vivo redox state of FBPase was not any more oxidized in ntrc trxy1y2 in comparison to the ntrc single mutant suggests that removal of Trx y’s likely has very little effect on the redox balance of 2-CysPrxs in vivo (assuming that the oxidizing effect on FBPase is caused via impaired reduction of 2-CysPrxs in these mutants). In any case, the authors should carry out an experiment to determine the in vivo redox state of 2-CysPrxs to clarify this issue.  

Author Response

  • The title of the paper is slightly misleading, and at least in the absence of further evidence should be modified to better reflect the actual findings. As it stands, there is rather little evidence of a functional relationship of any physiological significance between y-type Trxs and 2-CysPrxs.
    RESPONSE
    Title was changed to “Exploring the functional relationship between y-type thioredoxins and 2-Cys peroxiredoxins in Arabidopsis chloroplasts”
  • Line 20-21: Should say decreased 2-CysPrx amount partially rescues the phenotypes. Growth of those plants is still somewhat retarded in comparison to WT, as seen in a previous article from the author’s laboratory (ref [82]). It does seem to recover growth to the level of the 2cp mutant though.
    RESPONSE
    This sentence was modified in the abstract (line 22) and through the text (lines 87 and 214).
  • Please summarize the key findings of the study in addition to explaining the background and research question in the abstract.
    RESPONSE
    A summarizing sentence was added at the end of the Abstract (lines 26-29).
  • Lines 170–171: The Y(NPQ) parameter cannot be described as “non-regulated basal quenching”, that would better describe the Y(NO) parameter. Y(NPQ) is the yield of non-photochemical quenching (as correctly used in the Results and discussion section), which is very much a regulated, and a highly dynamic rather than basal process of thermal dissipation of excitation energy.
    RESPONSE
    Corrected (line 203).
  • Please add information about the age of plants used for protein alkylation assays and immunoblots, chlorophyll level determination, and Chl a fluorescence measurements in materials and methods in addition to the Figure legends.
    RESPONSE
    Information regarding experimental details was added to the methods section (lines 137-8, 178, 181-2, 194-5 and 197).
  • Lines 181-182: “While the direct interaction of NTRC with chloroplast redox-regulated enzymes cannot be ruled out” – the finding in ref. [26] that overexpression NTRC with an inactivated TRX domain still partially rescues the ntrc phenotype (without increasing the amount of reduced 2-CP) argues strongly in favour of this possibility. Also, I do not think that these two models for the influence pathway of NTRC are mutually exclusive. Both can be true at the same time in vivo.
    RESPONSE
    The sentence “While direct interaction…..” was deleted (lines 213-14) and the notion that NTRC interacts with Trxs in vivo is now mentioned in the Introduction (lines 78-80)
  • Figure 3 and Lines 273-274: It should be mentioned in the text too that Y(II) and Y(NPQ) were measured during photosynthetic induction, and not in a light curve like ETR. Also, what is the rational was using ETR in (A) and Y(II) in (B)? ETR is calculated from Y(II) by multiplying with PAR and the fraction of absorbed light and the ratio of PSII/PSI antennae, and I understand that the latter two multipliers were assumed constant between lines? (which is usually done even though there are often major differences in the ratio of PSII/PSI antenna sizes between mutant lines; this is especially true in mutants lacking NTRC, which have a significantly lowered amount of functional PSI: see PsaB protein level in ref. [29] and EPR and 77K spectra in Figures 3 & 4 in Nikkanen et al., 2019 Physiol Plantarum 166: 211-225). Therefore ETR offers really no relevant extra information compared to Y(II) here, while raising questions about the reliability of the values in the mutant lines. On top of this, In fact, using Y(II) also in (A) would make it possible to better distinguish between differences in the low light intensities, as the PAR value (which are of course same for all lines) in removed from the calculation.
    RESPONSE
    Text was added to clarify that Y(II) and Y(NPQ) were measured during photosynthetic induction (lines 322-3). Y(II) and Y(NPQ) figures were moved to 3A and B, respectively and the ETR figure was moved to the supplemental material (New S3). Text was modified accordingly (lines 326-335).
  • Table I: Why is Fv/Fm so low in WT? Healthy Col-0 plants should very reliably have Fv/Fm of c.a. 0.82-0.84. Where the plants stressed somehow? Also, are the t-tests of significance done in comparison to WT for all mutants? For the double mutants the relevant reference would in my view be the ntrc single mutant.
    RESPONSE
    Routinely, we observe Fv/Fm values lower than 0.82-0.83. Though we are not aware that plants are stressed, these lower Fv/Fm values might be an effect of the conditions in our
    plant growth chambers (temperature, humidity, etc). Nevertheless, the statistical analysis shows significant differences of Fv/Fm between the lines analyzed in this work.
  • Lines 284-285: ”..the additional deficiency of Trxs x and y impact photosynthetic parameters to a different extent”- although I suppose technically true, this is slightly misleading, because according to the data shown, additional deficiency of Trxs y in ntrc background has NO extra impact on photosynthetic parameters at all, while in ntrc trxx there is clear aggravation of the phenotype in comparison to ntrc.
    RESPONSE
    The sentence was rewritten to clarify that the deficiency of Trxs y does not aggravate photosynthetic parameters in NTRC-deficient plants (lines 342).
  • Line 343: There should be proper negative controls in the BiFC tests. Lack of interaction between Trx y2 and FBPase would work as a negative control for the Trx y2 side of the positive result, but it is not mentioned as such and does in fact seem to exhibit some YFP fluorescence. The authors should also demonstrate that the 2-CP constructs don’t interact with just everything. This is especially important since the YFP signal from Trx y2 + 2-CysPrxA in particular looks rather weak. In fact, it doesn’t look any stronger than the Trx y2 + FBPase test in Fig S3B, which is interpreted as a negative result.
    RESPONSE
    We have modified the BiFC results as follows:
    -We agree with the reviewer that the YFP signal from the Trxy2/2-Cys Prxs interaction is weak, which is now mentioned in the text (line 451-2). Hence, higher magnification images of BiFC analysis of the positive interactions between Trxy2 and 2-Cys Prxs (Fig 5A) and Trx x and 2-Cys Prxs (new Fig S4) are now included.
    - The BiFC analysis of Trxy2-FBPase interaction was moved to Fig S5A. For the 2-Cys Prxs negative control, we tested the interaction of these enzymes with chloroplast glutamine synthetase 2 (GS2) (new Fig S5B, lines 464-469), which is not a 2-Cys Prx interacting partner (Cerveau et al 2016). The NTRC-GS2 interaction, previously reported (Gonzalez et al., 2019), served as a positive control for GS2 (new Fig. S5B). The reference Cerveau et al 2016 was added ([52]).
    - Text was added (lines 454-458) to make clearer that FBPase and GS2 assays serve as negative controls.
  • The result that the in vivo redox state of FBPase was not any more oxidized in ntrc trxy1y2 in comparison to the ntrc single mutant suggests that removal of Trx y’s likely has very little effect on the redox balance of 2-CysPrxs in vivo (assuming that the oxidizing effect on FBPase is caused via impaired reduction of 2-CysPrxs in these mutants). The authors should carry out an experiment to determine the in vivo redox state of 2-CysPrxs to clarify this issue.
    RESPONSE
    The in vivo redox state of 2-Cys Prxs under different conditions (darkness, growth- and high light) was determined by MMPEG-based alkylation assays in long-day grown plants (new Figure 6A, B), and corresponding text (lines 524-581) was added. The results showed that, regardless of the light condition, the redox state of 2-Cys Prxs was similar in all tested lines except the ntrc-trxx mutant, which showed slightly more oxidized state of these enzymes were. We think that these findings are consistent with the notion that NTRC is the main reductant of 2-Cys Prxs, the redox state of which control the reducing capacity of
    chloroplast Trxs, thus the reductive activation of their targets. In the absence of NTRC, oxidized 2-Cys Prxs provoke drainage of reducing power from Trxs and, consequently, the redox regulation of Trx targets such as FBPase is impaired (Figure 4, refs 26, 29-31, 37). Therefore, it is not expected a high variation of the redox status of 2-Cys Prxs in ntrc mutants, as previously reported (Ojeda et al 2018 PCP, Guinea Diaz et al 2020 Plant Journal). Nevertheless, previous work from our group (Perez-Ruiz et al 2017) and other laboratories (Nikkanen et al 2016 PCE) have shown higher levels of 2-Cys Prxs oxidation in dark-adapted leaves of the ntrc mutant. A likely possibility to explain the variation of the redox state of 2-Cys Prxs in the ntrc mutant is that the leaf phenotype of this mutant is rather variable, young leaves showing more severe effects than older ones. We are now aware of this fact and perform analyses on complete rosettes. These contradictory observations and possible explanations are discussed in the text (lines 563-571). The references Dietz 2011 and Guinea et al 2020 were added ([53 and 54]).

Reviewer 2 Report

In this manuscript, Jurado-Flores et al have explored the functional relationship between Trx-y, NTRC and 2-Cys Prx in Arabidopsis. I believe this is a nice piece of work which has been properly performed, according to the methodological standards in the field. The results show that the phenotype of the mutant trxy1y2-ntrc is not as “appealing” as the previously published trxx-ntrc mutant, showing just a rather subtle additive phenotype in vegetative growth (respect to the single ntrc mutant), but statistically significant according to the authors. 

While I find the manuscript interesting and informative, I have some suggestions and concerns that I hope will help to improve it:

Major points:

1. I miss a few lines in the Introduction describing the proposed mechanisms of action of NTRC / 2-Cys Prx system and its crosstalk with FTR-Trx. It is also not clear what the proposed/reported physiological roles of Trx-y are. Is there a phenotype and/or physiological role assigned to trxy mutants? Additionally, I feel that the Introduction it is not properly framed within the context of light/dark regulation in chloroplasts. In my opinion, these points would facilitate the comprehension of the manuscript to non-expert readers.

2. Given the subtle limitation in vegetative growth of the mutant trxy1y2-ntrc, with respect to the single mutant ntrc, I would suggest to soften the statement in line 206: “there is a significant decrease in growth rate” to: “there is a slight, but statistically significant, decrease in growth rate”, or something similar.

3. Authors do not discuss what might be the physiological role (if any) of the interaction between Trx-y and 2-Cys Prx? Based on the results, it seems a weak interaction or, at least, weaker than the one between Trx-x and 2-Cys Prx.

4. Conclusions include a discussion of the main results of the manuscript. This is certainly interesting and appropriate but not in the “Conclusions” section. I’d suggest to move this to the end of the “Results and Discussion” section.

Figures:

.- Figure 5 is difficult to see. Do 2-Cys Prx and/or Trx-y2 show a similar speckle pattern?

.- Symbols in figure 2 cannot be properly distinguished.

Minor points:

.- Line 125: Please, check: Hi-Trap or His-Trap?

.- Line 379: the fact that “type?” Trx-y2

.- Abbreviations are sometimes not properly indicated (Trx, for instance). “Fdx” is stated in the abstract (line 14) but “Fd” is stated at the beginning of the Introduction (line 41)

.- I’d suggest to remove the adjectives “classical” and “typical” Trx, stated in lines 44 and 45 in the Introduction. Their meaning is confusing.

Author Response

  • I miss a few lines in the Introduction describing the proposed mechanisms of action of NTRC / 2-Cys Prx system and its crosstalk with FTR-Trx. It is also not clear what the proposed/reported physiological roles of Trx-y are. Is there a phenotype and/or physiological role assigned to trxy mutants? Additionally, I feel that the Introduction it is not properly framed within the context of light/dark regulation in chloroplasts. In my opinion, these points would facilitate the comprehension of the manuscript to non-expert readers.
    RESPONSE
    Text was added to describe the proposed relationship between the NTRC/2-Cys Prx and the FTR/Trx systems within the context of light/dark regulation in chloroplasts (Introduction, lines 82-92). Mentions to the physiological roles proposed for Trxs y are also included (Introduction, lines 99-105).
  • Given the subtle limitation in vegetative growth of the mutant trxy1y2-ntrc, with respect to the single mutant ntrc, I would suggest to soften the statement in line 206: “there is a significant decrease in growth rate” to: “there is a slight, but statistically significant, decrease in growth rate”, or something similar.
    RESPONSE
    The notion that growth rate was slightly aggravated in the ntrc-trxy1y2 double mutant was clarified through the text (lines 111-2, 238, 247 and 372).
  • Authors do not discuss what might be the physiological role (if any) of the interaction between Trx-y and 2-Cys Prx? Based on the results, it seems a weak interaction or, at least, weaker than the one between Trx-x and 2-Cys Prx.
    RESPONSE
    Although most plastidial Trxs reduce 2-Cys Prxs in vitro, NTRC is the primary reductant of these enzymes in vivo. Thereby, depletion of NTRC would increase drainage of reducing equivalents from TRXs to 2-Cys PRXs, impairing redox regulation of their targets (lines 82-92). Our results show that 2-Cys Prxs interact with Trx y2 in vitro and in planta, however, the comparative analysis (growth rate, photosynthesis performance, redox regulation of FBPase, oxidative stress response) of the ntrc-trxy1y2 and the ntrc and ntrc-trxx mutants suggests that the interaction of Trx y2 and 2 Cys Prxs is not physiologically relevant, at
    least under the conditions tested in this work. This is discussed in the results (lines 571-581) and the conclusion (lines 632-640).
  • Conclusions include a discussion of the main results of the manuscript. This is certainly interesting and appropriate but not in the “Conclusions” section. I’d suggest to move this to the end of the “Results and Discussion” section.
    RESPONSE
    The referred sentences (lines 626-630) were deleted.
  • Figure 5 is difficult to see. Do 2-Cys Prx and/or Trx-y2 show a similar speckle pattern?.
    RESPONSE
    We have replaced previous BiFC figures by higher magnification images (New Figures 3A, and S3). The speckle pattern shown by Trx y2 and 2-Cys Prxs and Trx x and 2-Cys Prxs is rather similar (lines 452-454).
  • Symbols in figure 2 cannot be properly distinguished.
    RESPONSE
    Symbols on Figure 2 were enlarged.
  • Line 125: Please, check: Hi-Trap or His-Trap?
    RESPONSE
    His-Trap, corrected (line 152)
  • Line 379: the fact that “type?” Trx-y2
    RESPONSE
    Previous sentence (482-484) was rewritten (lines 524-525).
  • Abbreviations are sometimes not properly indicated (Trx, for instance). “Fdx” is stated in the abstract (line 14) but “Fd” is stated at the beginning of the Introduction (line 41)
    RESPONSE
    Abbreviations for Trx was included (line 47) and Fdx was corrected (lines 46 and 47)
  • I’d suggest to remove the adjectives “classical” and “typical” Trx, stated in lines 44 and 45 in the Introduction. Their meaning is confusing.
    RESPONSE
    We removed “classical” and “typical” as requested (lines 30, 49, and 50)